# Morphology and Molecular Phylogeny Reveal Five New Species of *Laccaria* (Hydnangiaceae, Agaricales) from Southern China

**DOI:** 10.3390/jof9121179

**Published:** 2023-12-08

**Authors:** Ming Zhang, Xue-Lian Gao, Li-Qin Mu, Wang-Qiu Deng

**Affiliations:** 1Guangdong Provincial Key Laboratory of Microbial Culture Collection and Application, State Key Laboratory of Applied Microbiology Southern China, Institute of Microbiology, Guangdong Academy of Sciences, Guangzhou 510070, China; lajiawon@126.com; 2Chuxiong Yi Autonomous Prefecture Forestry and Grassland Science Research Institute, Chuxiong 675000, China

**Keywords:** fungal diversity, molecular systematics, morphology, novel taxa, taxonomy

## Abstract

The genus *Laccaria* is a type of cosmopolitan and ecologically important fungal group. Members can form ectomycorrhizal associations with numerous trees, and some species are common edible fungi in local markets. Although some new species from China are recently published, the species diversity of *Laccaria* is still unclear in China. In this study, some samples of *Laccaria* were collected from southern China, and morphological characteristics and phylogenetic analyses based on the multilocus dataset of ITS-LSU-*tef*1-*rpb*2 confirmed five new species. *Laccaria miniata*, *L. nanlingensis* and *L. neovinaceoavellanea* were collected from subtropical broad-leaved forests, and *L. rufobrunnea* and *L. umbilicata* were collected from subtropical mixed forests of southwest China. Full descriptions, illustrations, comparisons with similar species and phylogenetic analysis are provided.

## 1. Introduction

*Laccaria* Berk. & Broome (Agaricales, Hydnangiaceae) is an important fungal group with high ecological and economic values. Species in *Laccaria* are widely reported from every continent except Antarctica and can form ectomycorrhizal (ECM) relationships with a wide range of trees, such as Betulaceae, Fagaceae, Myrtaceae, Pinaceae and Salicaceae [1,2]. *Laccaria* is considered a model genus for understanding ectomycorrhizal (ECM) ecology and evolution; some species are reported that can act as pioneer species and can be frequently found in recently disturbed sites and young forest stands, play important roles in primary and secondary succession, and are useful for ecological protection and restoration [1,3,4,5,6,7,8,9,10,11]. In addition, some *Laccaria* species are edible and common in local markets [12]; for example, *L. aurantia*, *L. bicolor*, *L. moshuijun* and *L. vinaceoavellanea* are commercial mushrooms in Sichuan Province, China [13].

In general, *Laccaria* members are mainly characterized by the collybioid to omphaloid basidiocarp; the convex to plane or umbilicate, and usually dry pileus; thick, waxy, sinuate to subdecurrent, often widely spaced lamellae, echinulate basidiospores, the hymenium typically devoid of pleurocystidia, and the clamped hyphae [8,14,15,16,17,18]. It is not difficult to recognize the genus *Laccaria* due to its distinct morphological characteristics; however, infrageneric species identification is difficult in many instances due to the overlapping morphological features [18,19,20]. Recent studies show that molecular data are helpful in species identification of this genus [1,2,17,18,20,21,22]. The discovery of *Laccaria* new species has been rapidly increasing in recent years [23]. To date, about 110 species have been reported worldwide [2,23], but most of them were originally described from Europe and North America [1,8,20,24,25,26,27,28,29]. In China, 25 *Laccaria* species have been reported, including 17 species originally described from China [1,2,17,22,30,31,32,33,34]. 

During the survey of macrofungi in southern and southwestern China, some distinct *Laccaria* samples were collected. Based on morphological feature studies and multilocus phylogenetic analyses, five species are confirmed as new to science. Therefore, they are formally introduced herein, containing the full morphological descriptions, color photographs, line drawings, comparison with similar species, and a phylogenetic tree to show their placement and uniqueness.

## 2. Materials and Methods

### 2.1. Morphological Studies

Photographs of fresh basidiocarps were taken in the field. Specimens were dried and deposited in the Fungarium of Guangdong Institute of Microbiology (GDGM). Descriptions of macro-morphological characters and habitats were obtained from photographs and field notes. Color codes follow Kornerup and Wanscher [35]. Microscopic observations were carried out on tissue sections stained with 5% KOH and 1% aqueous Congo red under a light microscope (Olympus BX51, Tokyo, Japan) with magnification up to 1000×. For basidiospore descriptions, the abbreviation [n/m/p] denotes n spores measured from m basidiomata of p collections; the notation (a–)b–c(–d) describes basidiospore dimensions, where the range b–c represented 90% or more of the measured values and ‘a’ and ‘d’ were the extreme values; ‘av.’ represents the mean range of basidiospore length × width. Q referred to the length/width ratio of an individual basidiospore, and Q_m_ referred to the average Q value of all basidiospores ± sample standard deviation. All microstructure line drawings were made based on rehydrated materials.

Scanning electron microscopy (SEM) was applied to observe the surface of basidiospores. Lamellar fragments of the dried specimens were fastened to aluminum stubs and coated with gold palladium. Then, they were observed using a field emission scanning electron microscopy (Thermo Scientific Apreo 2S HiVac, Brno, Czech Republic) with an accelerating rate of 5 kV.

### 2.2. DNA Extraction, PCR Amplification and Sequencing

Genomic DNA samples were extracted from voucher specimens using the Sangon Fungus Genomic DNA Extraction Kit (Sangon Biotech Co., Ltd., Shanghai, China), according to the manufacturer’s instructions. Primer pairs ITS1F/ITS4 [36], LR0R/LR7 [37], EF1-983F/EF1-1967R [38] and bRPB2-6F/bRPB2-7.1R [39] were used to amplify ITS, LSU, *tef1* and *rpb2*, respectively. PCR reactions were performed in a total volume of 25 μL containing 0.5 μL template DNA, 11 μL sterile deionized water, 0.5 μL of each primer and 12.5 μL 2× PCR mix (DreamTaq^tm^ Green PCR Master Mix, Fermentas, MA, USA). Amplification reactions were performed in a TProfessional Standard Thermocycler (Biometra, Göttingen, Germany) under the following conditions: 95 °C for 4 min; then 35 cycles of denaturation at 94 °C for 60 s, annealing at 53 °C (ITS, LSU)/50 °C (*tef*1)/52 °C (*rpb*2) for 60 s, extension at 72 °C for 60 s and a final extension at 72 °C for 8 min. The PCR products were electrophoresed on 1% agarose gels and then sent for sequencing on an ABI Prism^®^ 3730 Genetic Analyzer (PE Applied Biosystems, Foster, CA, USA) at the Beijing Genomic Institute (BGI) using the same PCR primers. The raw sequences were assembled and checked with SeqMan implemented in Lasergene v7.1 (DNASTAR Inc., Madison, WI, USA). The newly generated sequences in this study were submitted to GenBank.

### 2.3. Phylogenetic Analyses

Sequences generated in this study and those downloaded from GenBank were combined and used for phylogenetic reconstruction. Detailed information of specimens included in this study was given in Table 1. Sequence matrices of ITS, LSU, *tef*1 and *rpb*2 were aligned separately with software MAFFT v7 using the E-INS-i strategy [40] and manually adjusted in MEGA 6 [41]. The ambiguously aligned regions and introns of the two protein-coding genes of *tef1* and *rpb2* were retained in the final analyses. Phylogenetic analyses were performed in PhyloSuite [42]. Maximum likelihood phylogenies were inferred using IQ-TREE [43] under the Edge-linked partition model (TPM2u+F+R4 for ITS and *rpb*2, TN+F+R3 for LSU, and TIM2e+G4 for *tef*1) for 5000 ultrafast bootstraps [44], as well as the Shimodaira–Hasegawa-like approximate likelihood-ratio test [45]. Bayesian Inference (BI) phylogenies were inferred using MrBayes 3.2.6 [46], and the best models of the multilocus datasets were searched via PartitionFinder 2 [47] for each locus, i.e., GTR+F+I+G4 for ITS and *tef*1, K80 + I + G for LSU, and SYM + I + G4 for *rpb*2. BI analysis using four chains was conducted by setting generations to 200,000 and the stoprul command with the value of stopval set to 0.01; trees were sampled every 1000 generations, the initial 25% of sampled data were discarded as burn-in and posterior probabilities (PP) were then calculated from the posterior distribution of the retained Bayesian trees. The phylogenetic trees were visualized in FigTree v1.4.23.

## 3. Results

### 3.1. Molecular Phylogeny

The combined dataset (ITS+LSU+*tef*1+*rpb*2) used for phylogenetic analyses consisted of 378 sequences from 185 collections, including 30 sequences (10 for ITS, 10 for nrLSU, 5 for *tef1* and 5 for *rpb2*) newly generated in the present study. The final alignment contained 2802 characters (704, 889, 614 and 595 for ITS, LSU, *tef1* and *rpb2,* respectively). *Mythicomyces corneipes* (Fr.) Redhead & A.H. Sm. was selected as the outgroup based on recent studies [2,33]. The phylogeny tree derived from the ML analysis with both PP and BS support values is shown in Figure 1. Phylogenetic analyses showed that *Laccaria* is a well-supported monophyletic group; specimens collected from China in the current study formed five highly supported monophyletic lineages within the genus. Three new species, *L. neovinaceoavellanea*, *L. rufobrunnea* and *L. umbilicata,* clustered together with *L. diospyricola*, *L. fengkaiensis, L. prave*, *L. vinaceoavellanea*, *L. violaceotincta* and *L. yunnanensis*, and formed a well-supported subclade. *Laccaria miniata* showed a close relationship with *L. glabripes*, *L. paraphysata* and *L. ohiensis*. Two specimens named *L. nanlingensis* formed a monophyletic clade.

### 3.2. Taxonomy

***Laccaria miniata*** Ming Zhang, sp. nov.; Figure 2a,b, Figure 3a,b and Figure 4.

Fungal Name: FN571672

Diagnosis—*Laccaria miniata* is distinguished by its small red basidiocarps with relatively longer stipes, longer basidia usually with two sterigmata (up to 11 μm long), globose to subglobose basidiospores, and the absence of cheilocystidia and caulocystidia. 

Etymology—‘*miniata*’ refers to the red basidiocarp.

Type—CHINA. Guangdong Province, Guangzhou City, Huadu District, Hongxiuquan Reservoir, elevation of 150 m, in a broad-leaved forest mainly dominated by Fagaceae trees, 23°27′ N, 113°12′ E, elevation of 100 m, 14 March 2019, Ming Zhang (GDGM76043), GenBank accession nos.: ITS = OR689440, 28S = OR785476.

Basidiocarps small, omphalinoid. Pileus 10–15 mm broad, convex to applanate, hygrophanous, glabrous to subtomentose, red (9A7–11A7), with a deep red (10C8–11C8) center, gradually fading to reddish orange to yellowish red (7A7–8A7) toward margin; margin entire, translucent-striate; context pale red (9A3–11A3). Lamellae sinuate to adnate, distant, pastel red (8A4–10A4), pastel pink (11A4) to red (9A6); lamellulae attenuate. Stipe 35–60 mm long, 1.5–2 mm thick, subcylindrical, subglabrous to fibrillose, sometimes slightly longitudinally striate, red (9A7–11A7) to deep red (10C8–11C8); basal mycelium white to grayish white (1A1–1A2). Odor and taste unknown.

Basidiospores (excluding ornamentation) [60/3/2] 8–10.5(11) × 8–10 μm, av. 9.37 ± 0.83 × 8.80 ± 0.60, Q = 1–1.11, Q_m_ = 1.06 ± 0.04, globose to subglobose, hyaline, echinulate, crowded; spines 0.5–1 μm long. Basidia 40–58 × 8–15 μm, clavate, hyaline, mostly two-spored, occasionally four-spored; sterigmata 6–11 μm long. Pleurocystidia and cheilocystidia not observed. Pileipellis, a cutis, composed of appressed, thin to slightly thick-walled (0.5 μm) filamentous hyphae 5–15 μm wide, colorless to slightly brownish. Lamellar trama regular, composed of thin-walled filamentous hyphae 5–12 μm wide. Stipitipellis is composed of appressed, parallel, simply septate, thin-walled, colorless hyphae 3–15 μm wide. Caulocystidia lacking. Clamp connections present.

Habitat and distribution—Single, scattered or in groups on soil in subtropical broad-leaved forests dominated by Fagaceae (*Castanopsis fissa*, *C.* spp.) trees. Currently known to be from southern China.

Notes—*Laccaria miniata* is mainly characterized by its tiny pileus, relatively slender stipe (stipe length 3–5 times that of pileus diameter), and the absence of pleurocystidia and cheilocystidia. On the base of the morphological features given above, the new species can be placed either in sect. *Laccata* or in sect. *Bisporae* of *Laccaria* [29].

*Laccaria pumila* Fayod, originally reported from France, is similar to *L. miniata* for having small basidiocarps; however, the former differs by its red-brown to orange-brown pileus usually fading to buff, pinkish flesh lamellae, interwoven pileipellis hyphae, larger basidiospores measuring (10)11–16.5(20) × (7.8)10–14.5(16) µm, and the north temperate habitations [55]. *Laccaria laccata* (Scop.) Cooke differs by the relatively larger basidiocarps, broadly elliptical basidiospores (8.5–9.5 × 6.7–8 μm), and the north temperate habitations [63,64]. *Laccaria longipes* G.M. Muell., originally reported from North America, also has long stipes; however, *L. longipes* differs by its larger basidiocarp, orange-brown to buff pileus, and subglobose to broadly elliptical basidiospores (av. 7.6–7.8 × 6.8–7.2 μm in size) [63]. 

Phylogenetic analysis shows *L. miniata* is close to *L. glabripes* McNabb*, L. ohiensis* (Mont.) Singer and *L. paraphysata* (McNabb) J.A. Cooper*. Laccaria glabripes*, originally reported from New Zealand, differs from *L. miniata* by its flesh pink to reddish brown pileus with a darker center, more robust stipe, and shorter four-spored basidia (32–48 × 7–10.5 μm) [65]; *Laccaria ohiensis*, originally reported from North America, differs by its relatively larger basidiocarp (pileus up to 35–50 mm in diameter), reddish brown to dark reddish brown pileus with finely furfuraceous tomentum on surface, larger basidiospores (9.5–12.5 µm in diameter) and two-spored basidia [59,65]; *L. paraphysata* differs by its relatively larger basidiocarp (pileus up to 35 mm broad), reddish brown to dark reddish brown pileus, irregularly shaped and simple branched paraphyses, and the presenting of cheilocystidia [65]. In addition, *L. paraphysata is* currently only known to be from New Zealand and grows under native bush and scrub dominated by *Leptospermum* spp. [65].

***Laccaria nanlingensis*** Ming Zhang, sp. nov.; Figure 2c,d, Figure 3c and Figure 5.

Fungal Name: FN571673

Etymology—‘nanlingensis’ refers to the locality of the type species in Nanling National Nature Reserve.

Diagnosis—*Laccaria nanlingensis* is characterized by its relatively larger basidiocarp, orange to brownish red pileus, pale red to grayish red lamellae, and small basidiospores (6.5–7.5 × 6–7 μm).

Type—CHINA. Guangdong Province, Shaoguan City, Nanling National Nature Reserve, 24°56′ N, 113°3′ E, elevation of 1000 m, 27 March 2021, Ming Zhang (GDGM84954), GenBank accession nos.: ITS = OR689442, 28S = OR785478, *tef*1 = OR826273, *rpb*2 = OR835199.

Basidiocarps small to medium, omphalinoid. Pileus 30–55 mm broad, convex to applanate, with a shallowly depressed center, dry or hygrophanous, glabrous, orange, reddish orange, orange-red, brownish orange to brownish red (5A6–8A6, 6C6–8C6), slightly fading to light orange to pale orange when dry, with obviously radial translucent striate, especially toward margin; margin entire, involute when young, applanate to wavy when old; context pinkish white to purplish white (13A2–14A2). Lamellae sinuate to adnate, distant, concolorous with pileus or darker to pale red to grayish red (9A4–10A4, 9C4–10C4); lamellulae attenuate. Stipe 25–70 mm long, 2–5 mm thick, cylindrical, subglabrous, occasionally slightly longitudinally striate, concolorous with pileus; basal mycelium white. Odor and taste unknown. 

Basidiospores (excluding ornamentation) [60/3/2] 6.5–7.5 × 6–7 μm, av. 6.95 ± 0.15 × 6.37 ± 0.45, Q = 1–1.16, Q_m_ = 1.09 ± 0.07, globose to subglobose, hyaline, echinulate, not crowded, distant; spines 0.5–1 μm long. Basidia 35–48 × 8–12 μm, clavate, hyaline, four-spored; sterigmata 6–10 μm long. Pleurocystidia lacking. Cheilocystidia 40–60 × 4–6 μm, filamentous to narrowly clavate, thin-walled, colorless and hyaline, abundant. Pileipellis a cutis with repent hyphae, thin to slightly thick-walled (0.5 μm) filamentous hyphae 5–13 μm wide, colorless to slightly brownish. Lamellar trama regular, composed of thin-walled filamentous hyphae 5–10 μm wide. Stipitipellis is composed of appressed, parallel, simply septate, thin to slightly thick-walled (0.5 μm), colorless to yellow-brown hyphae 4–8 μm wide. Caulocystidia 50–67 × 7–9 μm, clavate, scarce, sometimes subcapitate to irregularly shaped, slightly thick-walled (0.5 μm), colorless to yellow-brown, scattered. Clamp connections present.

Habitat and distribution—Single, scattered or in groups on soil in subtropical broad-leaved forests mainly dominated by Fagaceae trees. Currently known to be from southern China. 

Additional specimens examined—China. Guangdong Province, Shaoguan City, Ruyuan County, Nanling National Nature Reserve, 24°56′ N, 113°3′ E, elevation of 1000 m, 27 March 2017, Ming Zhang (GDGM84949), GenBank accession nos.: ITS = OR689441, 28S = OR785477, *tef*1 = OR826274, *rpb*2 = OR835198.

Notes—*Laccaria fagacicola* Yang-Yang Cui, Qing Cai & Zhu L. Yang, *L. himalayensis* A.W. Wilson & G.M. Muell. and *L. yunnanensis* Popa, Rexer, Donges, Zhu L. Yang & G. Kost are similar to *L. nanlingensis*. However, *L. fagacicola* differs from *L. nanlingensis* by its brownish orange to brownish pileus, relatively larger basidiospores (7–9 × 6.5–8 μm) and basidia (45–60 × 9–12 μm) [2]. *Laccaria himalayensis* differs by its larger basidiospores (av. 8.1–9.0 μm) and growing in mixed-temperate alpine conifer forests with *Abies*, *Acer*, *Larix*, *Pinus* and *Salix* [1]. *Laccaria yunnanensis* differs in having larger basidiocarps, brownish to flesh-colored pileus, flesh-colored lamellae, relatively larger basidiospores (8–9 × 8–10 μm), and the presence of pleurocystidia (55–65 ×15–25 μm) [22]. In addition, *L. torosa* H.J. Cho & Y.W. Lim also resembles *L. nanlingensis*; however, *L. torosa*, reported from Korea, differs in having larger basidiocarps, orange-brown to brown pileus fading to pale orange buff when dry or old, and larger basidiospores (8–9×8–9.5 μm) [50].

Phylogenetic analysis (Figure 1) showed that two specimens labeled as *L. nanlingensis* clustered together as an independent clade in the genus *Laccaria*, although its relationship to other *Laccaria* species is unclear.

Based on the morphological features, such as the dry or hygrophanous and reddish orange to brownish red pileus, the brownish red to grayish red lamellae, globose basidiospores with Q = 1–1.16, and the four-spored basidia, *L. nanlingensis* can be placed in the *Laccaria* sect. *Laccata* [29].

***Laccaria neovinaceoavellanea*** Ming Zhang & X.L. Gao, sp. nov.; Figure 2i, Figure 3d,e and Figure 6.

Fungal Name: FN571685

Diagnosis—*Laccaria neovinaceoavellanea* is distinctive by its pastel pink to pale violet pileus with a depressed center, globose to subglobose basidiospores with spines up to 2 μm long, and the presence of clavate, subcapitate to irregularly shaped caulocystidia.

Etymology—‘*neovinaceoavellanea*’ refers to the species similar to *L. vinaceoavellanea*. 

Type—CHINA. Jiangxi Province, Ganzhou City, Congyi County, Yangling National Forest Park, 25°37′ N, 114°19′ E, elevation of 400 m, 31 August 2016, Ming Zhang (GDGM52852), GenBank accession nos.: ITS = OR689447, 28S = OR785479.

Basidiocarps small, omphalinoid. Pileus 15–40 mm broad, convex to applanate, with a depressed center, dry or hygrophanous, subglabrous to subtomentosus, pastel pink, rose, purplish pink to pale violet (11A3–16A3) at mass, grayish magenta to dull violet (13D3–15D3) at center, with obviously radial translucent-striate, especially toward margin; margin entire, involute when young, applanate to wavy when old; context white to pinkish white or purplish white (13A2–14A2). Lamellae sinuate to adnate, distant, concolorous with pileus or paler; lamellulae attenuate. Stipe 30–70 mm long, 2–5 mm thick, cylindrical, subglabrous to fibrillose, occasionally slightly longitudinally striate, pastel red, rose to purplish pink (8A4–14A4); basal mycelium white. Odor and taste unknown. 

Basidiospores (excluding ornamentation) [30/2/2] 7–8 × 7–8 μm, av. = 7.6 ± 0.4 × 7.35 ± 0.42, Q = 1–1.14, Q_m_ = 1.04 ± 0.04, globose to subglobose, hyaline, echinulate, crowded; spines 1–2 μm long, 0.8–1.5 μm wide at base. Basidia 30–50 × 10–14 μm, clavate, hyaline, four-spored; sterigmata 4–6 μm long. Pleurocystidia lacking. Cheilocystidia 25–50 × 4–8 μm, filamentous to narrowly clavate, thin-walled, colorless and hyaline, abundant. Pileipellis a cutis with repent and occasionally suberect, thin to slightly thick-walled (0.5 μm) filamentous hyphae 6–18 μm wide, colorless to slightly brownish. Lamellar trama regular, composed of thin-walled filamentous hyphae 5–10 μm wide. Stipitipellis composed of appressed, parallel, simply septate, thin to slightly thick-walled (0.5 μm), colorless to yellow-brown hyphae 3–8 μm wide. Stipe trama composed of longitudinally arranged, infrequently branching, simple septate, thin-walled, colorless hyphae 3–8 μm wide. Caulocystidia 30–50 × 5–10 μm, clavate, sometimes subcapitate to irregularly shaped, slightly thick-walled (0.5 μm), colorless to yellow-brown, scattered. Clamp connections present. 

Habitat and distribution—Single, scattered or in groups on soil in subtropical broad-leaved forests mainly dominated by Fagaceae. Currently known to be from southwestern and southern China.

Additional specimens examined—China. Jiangxi Province, Ganzhou City, Congyi County, Yangling National Forest Park, 25°37′ N, 114°19′ E, elevation of 400 m, 2 September 2016, Ming Zhang (GDGM53063), GenBank accession nos.: ITS = OR689448, 28S = OR785480; Yunnan Province, Chuxiong Yi Autonomous Prefecture, Lufeng County, Guangtong Town, elevation of 2200 m, 29 August 2022, Xue-lian Gao 16 (GDGM89621), GenBank accession nos.: ITS = OR689449, 28S = OR785481.

Notes—*Laccaria amethystina* Cooke and *L. vinaceoavellanea* Hongo are morphologically similar to *L. neovinaceoavellanea*. *Laccaria amethystina* is different by an initially purple pileus gradually fading to buff or brownish, dark purple lamellae, finely to coarsely hairy or scaly stipe concolorous with pileus, and relatively larger basidiospores (7–10 μm in diameter) ornamented with longer spines (1.5–3 μm long) [8,28,66,67]. *Laccaria vinaceoavellanea* differs by a larger basidiocarp (pileus up to 60 mm broad), grayish buff, brownish vinaceous to vinaceous-buff pileus with furfuraceous at the margin, and relatively larger basidiospores (7.4–9.2 × 7.4–9.2 µm) [8,68]. 

Phylogenetic analysis (Figure 1) showed that *L. neovinaceoavellanea* is closely related to *L. vinaceoavellanea*, *L. violaceotincta* K.P.D. Latha, K.N.A. Raj & Manim. and *L. yunnanensis*. However, *L. violaceotincta*, originally reported from India, differs by its strongly hygrophanous and glabrous pileus, reddish gray or violet grays stipe covered with fine fibrils, and the presence of both two- and four-spored basidia [59]. *Laccaria yunnanensis* differs by its brownish to flesh-colored basidiocarp, flesh-colored lamellae, and larger basidiospores measuring (7.5)8–9 × (7.5)8–10 μm [22].

Based on the morphological features, such as the purplish pink to pale violet basidiocarps, globose basidiospores with Q = 1–1.14, four-spored basidia, and the colorless to slightly brownish pileipellis hyphae, *L. neovinaceoavellanea* can be placed in the *Laccaria* sect. *Violaceae* [29].

***Laccaria rufobrunnea*** Ming Zhang & X.L. Gao, sp. nov.; Figure 2e,f, Figure 3f and Figure 7.

Fungal Name: FN571674

Diagnosis—*Laccaria rufobrunnea* is distinctive in the genus *Laccaria* by its brownish orange to brownish red pileus, pastel red to purplish pink lamellae, white to pinkish white stipe, and the absence of caulocystidia. 

Etymology—‘*rufobrunnea*’ refers to the brownish-red pileus color. 

Type: China. Yunnan Province, Chuxiong Yi Autonomous Prefecture, Nanhua County, 25°15′N, 101°16′ E, elevation of 1900 m, 26 August 2020, Ming Zhang (GDGM82878), GenBank accession nos.: ITS = OR689443, 28S = OR785482, *tef*1 = OR826272, *rpb*2 = OR835197. 

Basidiocarps small, omphalinoid. Pileus 12–35 mm broad, convex to subapplanate, hygrophanous, glabrous, brownish orange to brownish red (6C7–9C7), with a darker center, usually fading to brownish yellow (5C7) in dry condition; margin entire, involute to incurved when young, incurved to decurved when old, obscurely translucent-striate; context brownish gray (5C2–8C2). Lamellae sinuate to adnate, distant, pastel red, pink to purplish pink (10A4–14A4), usually changing pinkish white to purplish white in dry condition or old (10A2–14A2); lamellulae attenuate. Stipe 20–40 mm long, 3–5 mm thick, cylindrical, subglabrous, occasionally slightly longitudinally striate, white to pinkish white (7A2–10A2); basal mycelium white. Odor and taste unknown. 

Basidiospores (excluding ornamentation) [60/3/2] 8–9 × 7–8 μm, av. 8.15 ± 0.23 × 7.57 ± 0.5, Q = 1–1.14, Q_m_ = 1.08 ± 0.06, globose to subglobose, hyaline, echinulate, not crowded, subdistant; spines 1–2 μm long. Basidia 40–54 × 10–15 μm, clavate, hyaline, four-spored; sterigmata 5–7 μm long. Pleurocystidia lacking. Cheilocystidia 35–50 × 3–7 μm, filamentous to narrowly clavate, thin-walled, colorless and hyaline, abundant. Pileipellis a cutis with repent hyphae, thin to slightly thick-walled (0.5 μm) filamentous hyphae 5–12 μm wide, colorless to slightly brownish. Lamellar trama regular, composed of filamentous hyphae 3–6 μm wide. Stipitipellis composed of appressed, parallel, simply septate, thin to slightly thick-walled (0.5 μm), colorless to yellow-brown hyphae 3–9 μm wide. Stipe trama composed of longitudinally arranged, infrequently branching, simple septate, thin-walled, colorless hyphae 3–9 μm wide. Caulocystidia lacking. Clamp connections present.

Habitat and distribution—Single, scattered or in groups on soil in subtropical mixed forests mainly dominated by Fagaceae trees (such as *Quercus yunnanensis*, *Q. variabilis*, *Castanopsis hystrix*, *Castanopsis* spp.) and pine trees (such as *Pinus khasys*, *Pinus yunnanensis*). Currently known to be from southwestern China.

Additional specimen examined—China. Yunnan Province, Chuxiong Yi Autonomous Prefecture, Lufeng County, Guangtong Town, elevation of 2200 m, 29 August 2022, Xue-lian Gao 26 (GDGM89627), GenBank accession nos.: ITS = OR689444, 28S = OR785483.

Notes—*Laccaria rufobrunnea* is characterized by its brownish orange to brownish red pileus, pastel red to purplish pink lamellae, white to pinkish white stipe, and the absence of caulocystidia. On the basis of the morphological features described above, *L. rufobrunnea* can be placed in *Laccaria* sect. *Violaceae* [29].

 Phylogenetic analysis supported *L. rufobrunnea* as a distinct lineage in *Laccaria*, and close to *L. prava* Fang Li and *L. umbilicata* Ming Zhang. However, *L. prava* differs by its larger basidiocarp (pileus up to 7.5 cm broad), reddish white to pastel red pileus with strongly striate or rugulose-sulcate, reddish white to grayish red lamellae, longer stipe and smaller basidiospores measuring 6.5–7.5 × 7–8 μm [33]; *L. umbilicata* differs by its pale orange to light orange pileus, orange white to pinkish white lamellae without purple tinge, broader basidiospores (8–10 μm in diameter) and the present of caulocystidia (present study).

***Laccaria umbilicata*** Ming Zhang, sp. nov.; Figure 2g,h, Figure 3g,h and Figure 8.

Fungal Name: FN 571682

Diagnosis—*Laccaria umbilicata* is characterized by a pale orange to light orange colored pileus, white to orange-white stipe, globose to subglobose basidiospores (7)8–10 × (7)8–10 μm, and the absence of pleurocystidia. 

Etymology—‘*umbilicata*’ refers to the umbilicate pileus.

Type—China. Yunnan Province, Chuxiong Yi Autonomous Prefecture, Nanhua County, 25°15′ N, 101°16′ E, elevation of 1990 m, 26 August 2020, Ming Zhang (GDGM82911), GenBank accession nos.: ITS = OR689446, 28S = OR785486, *tef*1 = OR826268, *rpb*2 = OR835192.

Basidiocarps small, omphalinoid. Pileus 10–28 mm broad, convex to applanate, a depressed center, dry or hygrophanous, glabrous, pale yellow, pale orange to light orange (4A3–6A3, 4A5–6A5), slightly changing light brown to brown (5D7–6D7) when in dry condition, especially toward margin; margin entire, involute when young, applanate when old, with obscurely translucent-striate; context white to orange white (5A1–5A2). Lamellae sinuate to adnate, distant, orange-white, to pinkish white (5A2–7A2), sometimes pale orange to pastel red in dry conditions (5A3–7A3); lamellulae attenuate. Stipe 20–40 mm long, 2–3 mm thick, cylindrical, subglabrous, occasionally slightly longitudinally striate, white, yellowish white to orange-white (4A2–5A2); basal mycelium white. Odor and taste unknown. 

Basidiospores (excluding ornamentation) [60/3/2] (7)8–10 × (7)8–10 μm, av. 9.03 ± 0.75 × 8.67 ± 0.82, Q = 1–1.125, Q_m_ = 1.04 ± 0.06, globose to subglobose, hyaline, echinulate, not crowded, subdistant; spines 1.5–3 μm long, up to 2 μm wide at base. Basidia 40–45 × 10–14 μm, clavate, hyaline, four-spored; sterigmata 6–10 μm long. Pleurocystidia lacking. Cheilocystidia 30–47 × 3–8 μm, filamentous to narrowly clavate, thin-walled, colorless and hyaline, abundant. Lamellar trama subregular, composed of thin-walled filamentous hyphae 3–6 μm wide. Pileipellis a cutis with repent and occasionally suberect, thin to slightly thick-walled (ca. 0.5–1 μm) filamentous hyphae 3–18 μm wide, colorless to slightly brownish. Subhymenium composed of filamentous hyphae 3–10 μm wide. Stipitipellis composed of appressed, parallel, simply septate, thin to slightly thick-walled (0.5 μm), colorless to yellow-brown hyphae 3–8 μm wide. Stipe trama composed of longitudinally arranged, infrequently branching, simple septate, thin-walled colorless hyphae 3–8 μm wide. Caulocystidia 28–38 × 4–6 μm, clavate, sometimes subcapitate, slightly thick-walled (0.5 μm), colorless to yellow-brown, scattered. Clamp connections present.

Habitat and distribution—Single, scattered or in groups on soil in subtropical mixed forests mainly dominated by *Pinus yunnanensis* Franchet, mixed with a small number of Fagaceae trees. Currently, it is only known to be from southwestern China.

Additional specimens examined—China. Yunnan Province, Chuxiong Yi Autonomous Prefecture, Nanhua County, 25°15′ N, 101°16′ E, elevation of 2000 m, 26 August 2020, Ming Zhang (GDGM82883), GenBank accession nos.: ITS = OR689445, 28S = OR785485, *tef*1 = OR826270, *rpb*2 = OR835194; same locality and data (GDGM82851), GenBank accession nos.: 28S = OR785484, *tef*1 = OR826271, *rpb*2 = OR835195.

Notes—*Laccaria umbilicata* can be easily distinguished from other *Laccaria* species by its pale orange to light orange pileus with a depressed center, orange-white to pinkish white lamellae, white to orange-white stipe, and the absence of pleurocystidia. Based on the morphological features described above, the new species can be placed in the *Laccaria* sect. *Laccata* [29].

*Laccaria alba* Zhu L. Yang & Lan Wang is morphologically similar to *L. umbilicata*; however, *L. alba* differs by its white to whitish basidiocarp, pinkish lamellae and smaller caulocystidia measuring 10–15 × 4–6 µm [30].

Phylogenetically, three specimens named *L. umbilicata* formed a distinct lineage in *Laccaria*, and sister to *L. prava*; however, *L. prava* can be easily distinguished by its larger basidiocarps (pileus up to 7.5 cm broad) and smaller basidiospores (6.5–7.5 × 7–8 μm) [33]. The two species together with *L. diospyricola*, *L. fengkaiensis*, *L. neovinaceoavellanea, L. rufobrunnea, L. violaceotincta, L. vinaceoavellanea* and *L. yunnanensis* formed a well-supported Asian distribution subclade (BS/BPP=100%/1), among which, *L. diospyricola* and *L. violaceotincta* originally reported from southwestern India, *L. vinaceoavellanea* reported from Japan, and the others from China.

## 4. Discussion

*Laccaria* is a monophyletic group in the family Hydnangiaceae [2,23,69]. Five species-level lineages representing five new species from China were uncovered in the *Laccaria* clade.

Previous studies based on morphological characteristics divided *Laccaria* into several different infrageneric classifications [14,29,70,71]. For example, Bon [70] and Ballero and Contu [71] divided the genus *Laccaria* into three sections (sect. *Maritimae*, sect. *Amethystinae* and sect. *Laccata*), and the section *Laccata* was further divided into several subgroups. Singer [14] recognized five stirps (stirp *Trullissata*, stirp *Amethystina*, stirp *Laccata*, stirp *Galerinoides*, and stirp *Purpureobadia*) in *Laccaria*. However, Pázmány [48] established two subgenera (subg. *Maritimae* and subg. *Laccata* ‘as *Laccaria*’) in the genus *Laccaria*, and the subg. *Laccata* was further divided into five sections (sect. *Bisporae*, sect. *Laccata*, sect. *Obscurae*, sect. *Purpureobadia* and sect. *Violaceae*). However, those systematic arrangements were often unstable, unnatural and contradictory, and were not supported by phylogenetic studies based on DNA sequences. Thus, the infrasubgeneric classification of *Laccaria* is still unclear, and more studies are needed to improve it.

Because the species in *Laccaria* are very similar in morphology, and the characters of some species change greatly in different growth stages and humidity conditions, there can be difficulties in terms of species identification. The basidiospore dimensions play a crucial role in species identification in most macrofungi groups, but in *Laccaria*, this characteristic may be confusing because some species share similar size and shape basidiospores. For example, the five new species described above all share with globose to subglobose basidiospores with a Q-value between 1 to 1.1 and the size of basidiospores between 7–10 μm in diameter, except *L. nanlingensis*. However, the length and density of spines on the spore surface can be a useful distinguishing feature; for example, *L. acanthospora*, *L. alba* and *L. angustilamella* have longer spines that can reach up to 5–6 μm long, while *L. miniata* and *L. prava* have shorter spines that are less than 1 μm long (see Table 2); *L. miniata*, *L. neovinaceoavellanea* and *L. umbilicata* have relatively more crowded spines than *L. nanlingensis,* and *L. rufobrunnea* (see Figure 3, Figure 4, Figure 5, Figure 6, Figure 7 and Figure 8). In the description of the above five species, the words ‘crowded’, ‘subdistant’, and ‘distant’ were used to distinguish the different densities of spines on basidiospores surfaces.

In addition, basidia size and whether they are two- or four-spored can be useful distinguishing features. Most species of *Laccaria* have four-spored basidia, while *L. echinospora, L. fraterna* and *L. nigra* have two-spored basidia, *L. aurantia,* and the new species *L. miniata* have both two- and four-spored basidia, which are infrequent in *Laccaria*. 

Additionally, ecological information such as host plant species and habitat are also important and can provide useful clues for species identification. *Laccaria miniata* and *L. nanlingensis* collected from subtropical broad-leaved forests of southern China and could be associated with trees of Fagaceae; *L. rufobrunnea* and *L. umbilicata* distributed in subtropical mixed forests (which mainly dominated by Fagaceae trees and pine trees) of southwest China; and *L. neovinaceoavellanea* can naturally distributed in subtropical broad-leaved forests of southern and southwestern China, and can form symbiotic relationship with Fagaceae trees. On the basis of our study, the main morphological characters and ecological information of each species in *Laccaria* described from China are summarized in Table 2. 

Although many species of *Laccaria* have been reported in China, most species are mainly reported from southwestern regions, and the species recognition of *Laccaria* in China is still in its infancy, especially in tropical and subtropical areas. In the past, many Chinese samples of *Laccaria* were inaccurately labeled as *L. laccata*, *L. bicolor*, *L. amethystea* and *L. vinaceoavellanea* [72,73,74,75]. Phylogenetic studies have made some species well-understood in China, and the discovery of new species in *Laccaria* is rapidly increasing [2,17,21,22,31,32,33,34,50]. However, the distribution of *L. amethystea*, *L. bicolor* and *L. laccata* in China still needs to be investigated. A broader taxon sampling coupled with both molecular and morphological data is needed to fully understand the species diversity of *Laccaria* in China.

## Figures and Tables

**Figure 1 jof-09-01179-f001:**
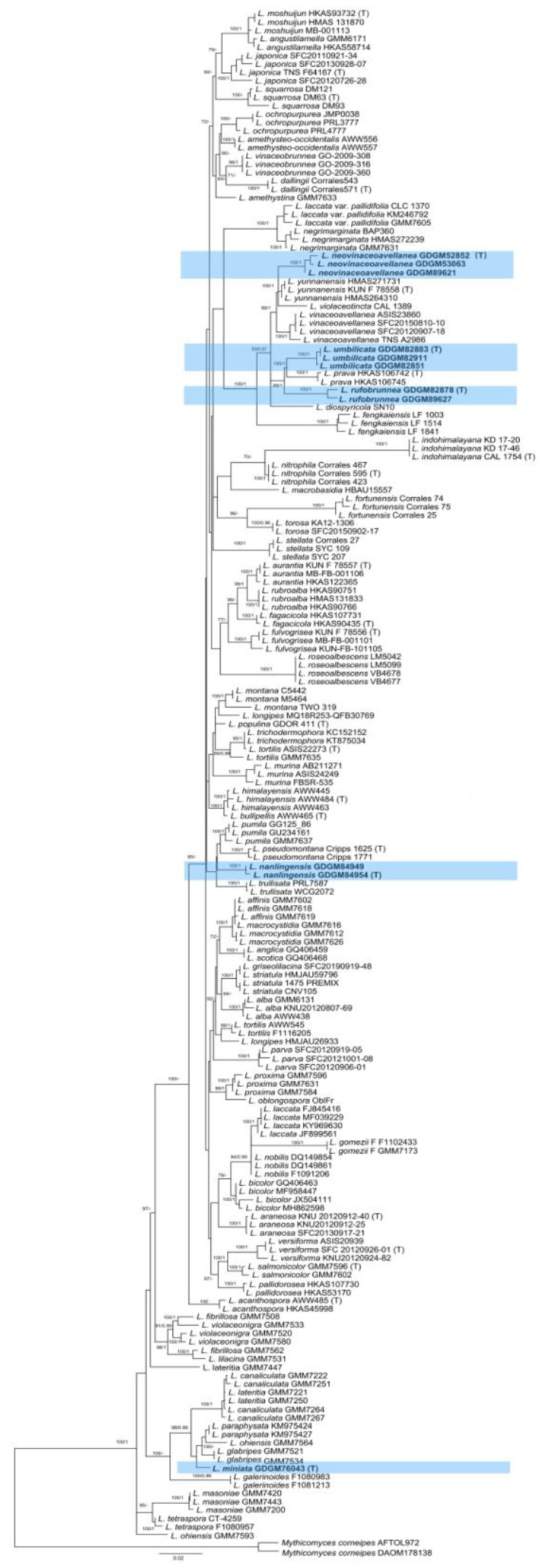
Maximum-likelihood phylogenetic tree of *Laccaria* generated from the ITS-LSU-*tef1*-*rpb2* dataset. Bootstrap values (ML ≥ 70%) and Bayesian posterior probabilities (BPP ≥ 0.95) are shown around branches. Sequences from type specimens are marked with (T), and the new species is indicated in bold and blue area.

**Figure 2 jof-09-01179-f002:**
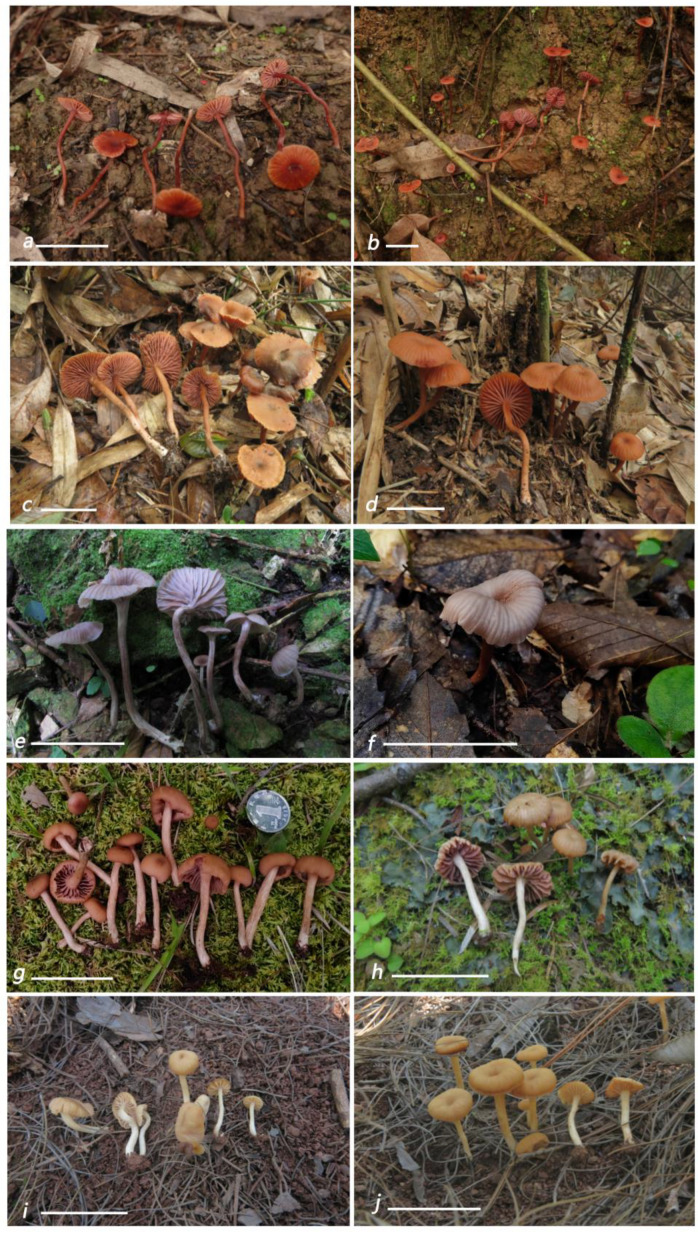
Fresh basidiomata of five new species of *Laccaria*. (**a**,**b**) *L. miniata* (Type, GDGM76043). (**c**,**d**) *L.* nanglingensis ((**c**). GDGM84949; (**d**). Type, GDGM84954). (**e**,**f**) *L. neovinaceoavellanea* ((**e**). Type, GDGM52852; (**f**). GDGM89621). (**g**,**h**) *L. rufobrunnea* ((**g**). Type, GDGM82787, (**h**). GDGM89627). (**i**,**j**) *L. umbilicata* ((**i**). Type GDGM82911, (**j**). GDGM82851). Bars: (**a**,**b**) = 20 mm, (**c**–**j**) = 50 mm.

**Figure 3 jof-09-01179-f003:**
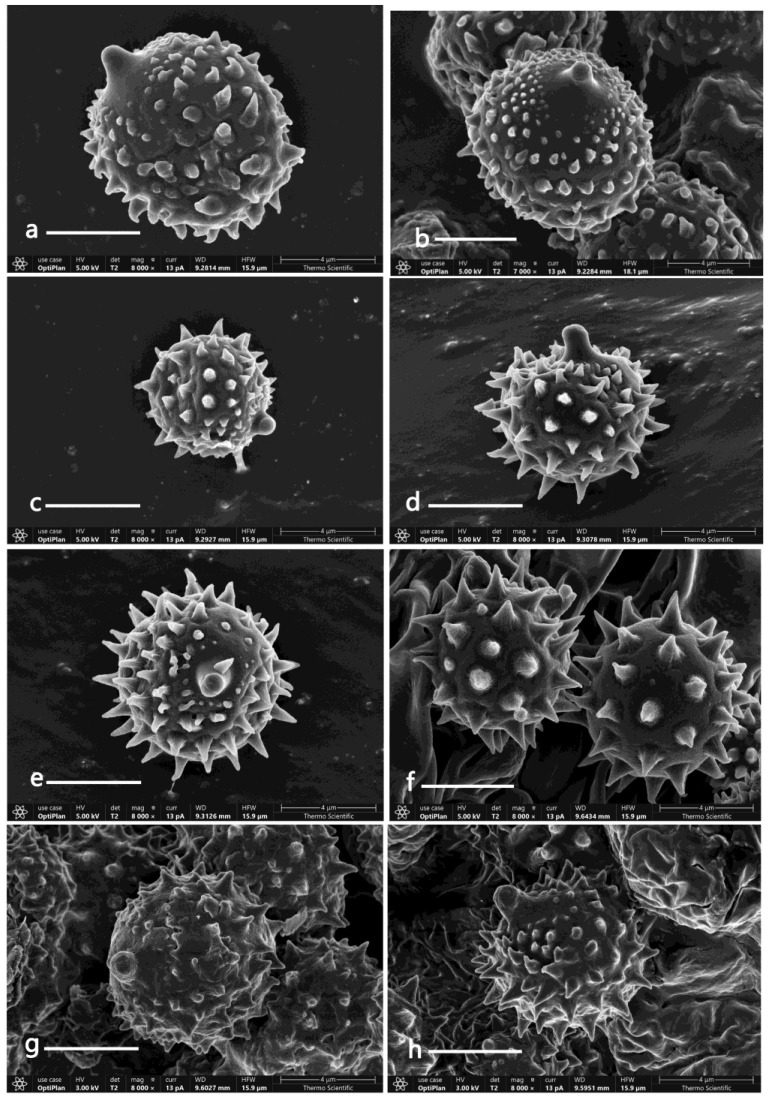
Basidiospores of the five new *Laccaria* species under SEM. (**a**,**b**) *L. miniata* (GDGM76043). (**c**) *L.* nanglingensis (GDGM84954). (**d**,**e**) *L. neovinaceoavellanea* (GDGM52852). (**f**) *L. rufobrunnea* (**f**. GDGM82787). (**g**,**h**) *L. umbilicata* (GDGM82911). Bars: (**a**–**h**) = 4 μm.

**Figure 4 jof-09-01179-f004:**
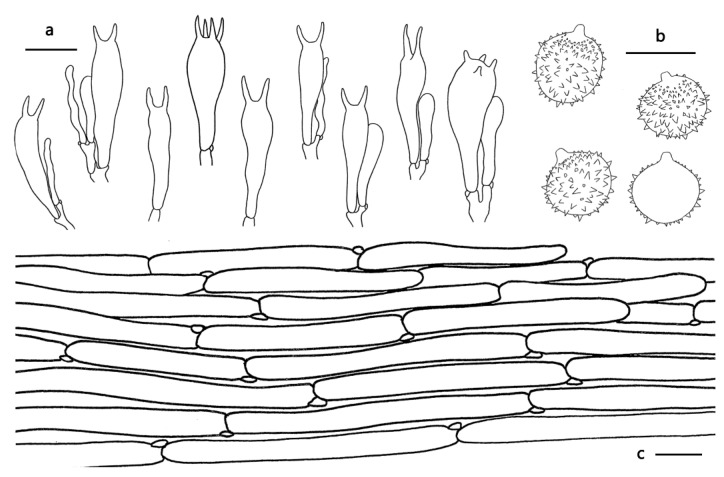
Microscopic features of *Laccaria miniata* (GDGM76043, Holotype). (**a**) Basidia. (**b**) Baisidiospores (**c**) Pileipellis. Bars: (**a**,**c**) = 20 μm, (**b**) = 10 μm.

**Figure 5 jof-09-01179-f005:**
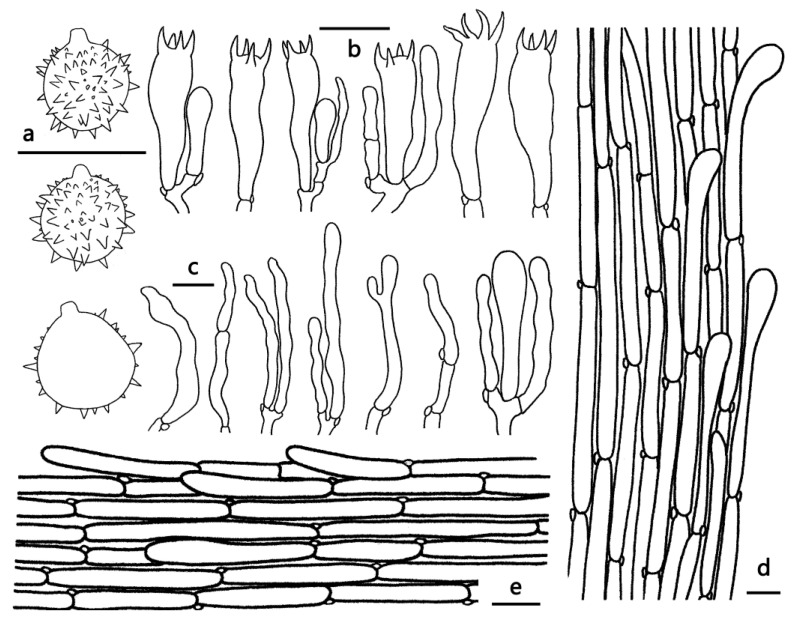
Microscopic features of *Laccaria nanlingensis* (GDGM84954, Holotype). (**a**) Basidiospores. (**b**) Basidia. (**c**) Cheilocystidia. (**d**) Stipitipellis and caulocystidia. (**e**) Pileipellis Bars: (**a**,**c**,**d**) = 10 μm; (**b**,**e**) = 20 μm.

**Figure 6 jof-09-01179-f006:**
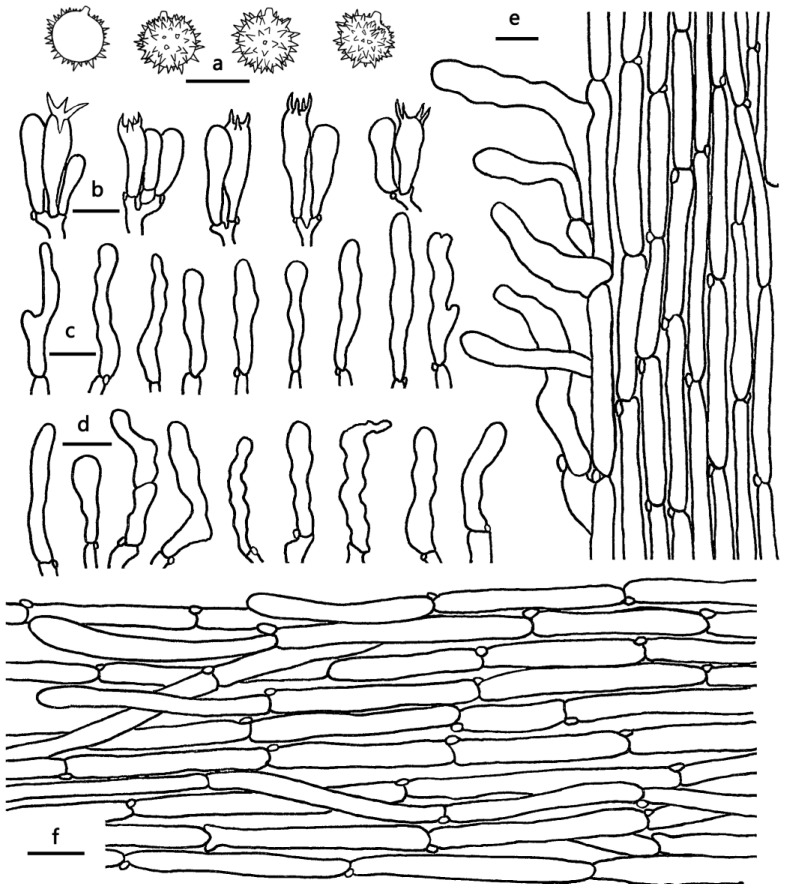
Microscopic features of *Laccaria neovinaceoavellanea* (GDGM52852, Holotype). (**a**) Basidispores. (**b**) Basidia. (**c**) Cheilocystidia. (**d**) Caulocystidia. (**e**) Pileipellis. (**f**) Stipitipellis. Bars: (**a**,**e**) = 10 μm; (**b**–**d**,**f**) = 20 μm.

**Figure 7 jof-09-01179-f007:**
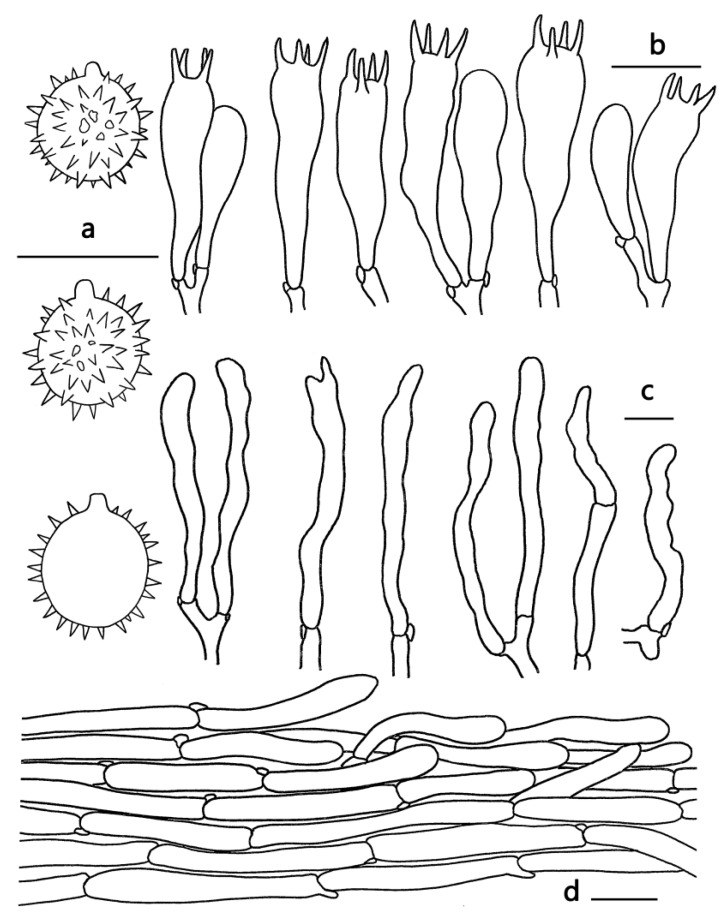
Microscopic features of *Laccaria rufobrunnea* (GDGM82878, Holotype). (**a**) Basidiospores. (**b**) Basidia. (**c**) Cheilocystidia; (**d**) Pileipellis. Bars: (**a**,**c**) = 10 μm; (**b**,**d**) = 20 μm.

**Figure 8 jof-09-01179-f008:**
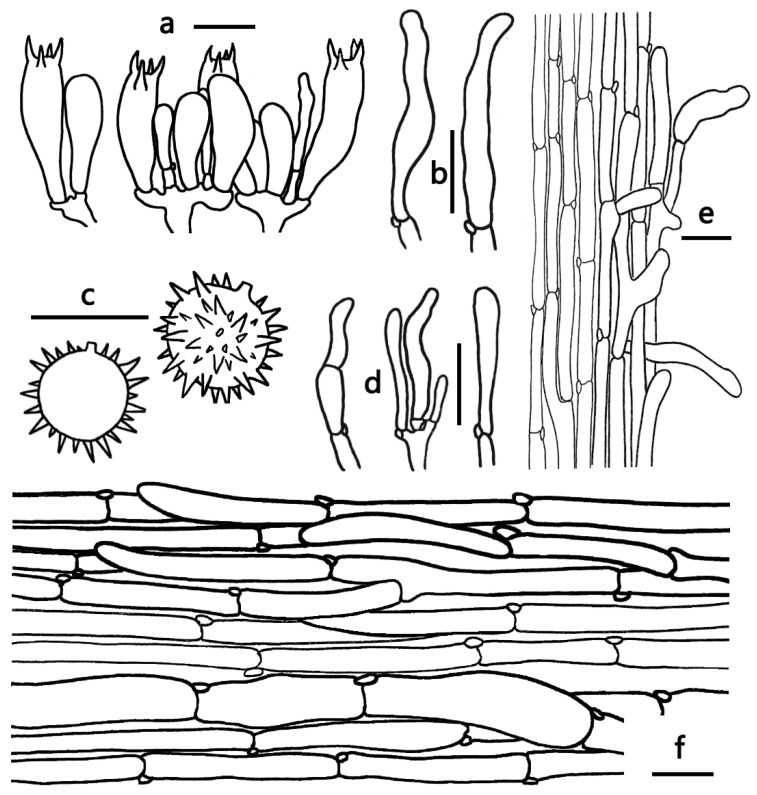
Microscopic features of *Laccaria umbilicata* (GDGM82911, Holotype). (**a**) Basidia. (**b**) Cheilocystidia. (**c**) Basidiospores. (**d**) Caulocystidia. (**e**) Stipitipellis and caulocystidia. (**f**) Pileipellis Bars: (**a**,**b**,**d**) = 20 μm; (**c**,**e**,**f**) = 10 μm.

**Table 1 jof-09-01179-t001:** Taxa included in molecular phylogenetic analyses and their GenBank accession numbers. Accession numbers in bold indicate newly generated sequences in this study. Taxa marked with T represent type specimens.

Taxa	Specimens	Locality	GenBank Accession Nos.				References
			ITS	LSU	*tef*1	*rpb*2	
*Laccaria acanthospora* (T)	AWW485	Tibet, China	JX504102	JX504186	KU686073	KU685916	[21]
*L. acanthospora*	HKAS45998	Tibet, China	JX504161	KU685870	_	KU686069	[1]
*L. affinis*	GMM7618	France	KM067852	_	_	_	[48]
*L. affinis*	GMM7619	France	KM067853	_	_	_	[48]
*L. affinis*	GMM7602	France	KM067842	_	_	_	[48]
*L. alba*	GMM6131	China	JX504131	JX504210	KU686079	KU685930	[21]
*L. alba*	KUN 20120807-69	China	MG519542	MG519583	MG551649	MG551616	[1]
*L. alba*	AWW438	China	JX504094	JX504178	KU686072	KU685912	[1]
*L. amethysteo-occidentalis*	AWW557	USA	MT279220	MT279200	MT436061	MT431174	[49]
*L. amethysteo-occidentalis*	AWW556	Canada	JX504107	JX504191	_	KU685919	[1]
*L. amethystina*	GMM7633	France	JX504154	JX504228	_	_	[21]
*L.* *angustilamella*	GMM6171	France	JX504132		_	_	[21]
*L.* *angustilamella*	HKAS58714	Tibet, China	JX504168	JX504244	_	_	[21]
*L.* *anglica*	AngFr	France	GQ406459	_	_	_	[49]
*L.* *anglica*	ScoFr	France	GQ406468	_	_	_	[49]
*L. araneosa* (T)	KNU20120912-40	Korea	MG519548	MG519588	MG551654	MG551621	[50]
*L.* *araneosa*	KUN20120912-25	Korea	MG519550	MG519590	MG551656	MG551623	[1]
*L.* *araneosa*	SFC20130917-21	Korea	MG519549	MG519589	_	MG551622	[1]
*L. aurantia* (T)	KUN-F78557	Yunnan, China	JQ670895	_	_	_	[22]
*L.* *aurantia*	HKAS122365	Yunnan, China	ON794252	_	_	_	Direct Submission
*L. aurantia*	MB-FB-001106	Yunnan, China	JQ670895	_	_	_	[22]
*L.* *bicolor*	BicSer	France	GQ406463	_	_	_	[49]
*L.* *bicolor*	LbC	United Kingdom	MF958447	_	_	_	[51]
*L.* *bicolor*	AWW585	USA	JX504111	_	_	_	[21]
*L.* *bicolor*	CBS:559.96	Netherlands	MH862598	_	_	_	[52]
*L. bullipellis* (T)	AWW465	Tibet, China	JX504100	JX504184	_	KU685914	[21]
*L. canaliculata*	GMM7222	Australia	KU685664	KU685807	_	KU685950	[1]
*L. canaliculata*	GMM7264	Australia	KU685674	KU685817	_	KU685957	[1]
*L. canaliculata*	GMM7267	Australia	JX504137	JX504213	KU686093	KU685960	[1]
*L. canaliculata*	GMM7251	Australia	KU685669	KU685812	KU686090	KU685955	[1]
*L. dallingii*	Corrales 543	Panama	MT279238	MT279213	MT436076	MT431187	[53]
*L. dallingii* (T)	Corrales 571	Panama	MT279240	MT279214	_	_	[53]
*L. diospyricola*	SN 10	India	MK776767	_	_	_	Direct Submission
*L. fagacicola* (T)	HKAS90435	Yunnan, China	MW540806	_	_	_	[2]
*L.* *fagacicola*	HKAS107731	Yunnan, China	MW540807	_	_	_	[2]
*L. fengkaiensis* (T)	HKAS106739	Guangdong, China	MN585657	_	_	_	[33]
*L.* *fengkaiensis*	HKAS106741	Guangdong, China	MN585658	_	_	_	[33]
*L.* *fengkaiensis*	LF 1841	China	MT822919	_	_	_	[54]
*L. fibrillosa*	GMM7508	New Zealand	KU685706	KU685847	_	KU685989	[1]
*L. fibrillosa*	GMM7562	New Zealand	KU685714	KU685855	_	KU685996	[1]
*L. fortunensis*	Corrales 74	Panama	MT279246	_	_	_	[53]
*L. fortunensis*	Corrales 75	Panama	MT279247	_	_	_	[53]
*L. fortunensis*	Corrales 25	Panama	MT279245	_	_	_	[53]
*L. fulvogrisea* (T)	KUN-F78556	Yunnan, China	JQ670896	_	_	_	[22]
*L.* *fulvogrisea*	KUN-FB-101105	Yunnan, China	JQ681210	_	_	_	[22]
*L.* *fulvogrisea*	MB-FB-001101	Yunnan, China	JQ670896	_	_	_	Direct Submission
*L. galerinoides*	F1081213	Chile	KU685634	KU685778	KU686078	KU685929	[1]
*L. galerinoides*	F1080983	Argentina	KU685632	KU685776	KU686077	KU685927	[1]
*L. gomezii*	F1102433	Costa Rica	_	MT279205	_	MT431180	[1]
*L. gomezii*	GMM7173	Costa Rica	MT279227	MT279207	MT436071	MT431182	[51]
*L. griseolilacina*	SFC20190919-48	South Korea	MT322981	MT322983	MT333269	MT333266	[23]
*L. himalayensis* (T)	AWW484	Tibet, China	JX504101	JX504185	_	KU685915	[21]
*L.* *himalayensis*	AWW445	Tibet, China	JX504096	JX504180	_	_	[21]
*L.* *himalayensis*	AWW463	Tibet, China	JX504098	JX504182	_	KU685913	[21]
*L. indo* *himalayana*	KD 17-46	India	MK575505	_	_	_	Direct Submission
*L. indohimalayana* (T)	KD 17-20	India	MK584157	_	_	_	Direct Submission
*L. indo* *himalayana*	CAL 1754	India	NR_171835	_	_	_	Direct Submission
*L. japonica* (T)	TNS-F64167	Honshu, Japan	KU962988	_	_	_	[32]
*L.* *japonica*	SFC20130704-34	Korea	MG519521	MG519568	MG551596	MG551598	[55]
*L.* *japonica*	SFC20130928-07	Japan	KU962975	MG519565	MG551632	MG551594	[55]
*L.* *japonica*	SFC20120726-28	Japan	MG519520	MG519567	MG551634	MG551597	[1]
*L. laccata*	SMI199	Canada	FJ845416	_	_	_	[56]
*L. laccata*	L16	Mexico	MF039229	_	_	_	Direct Submission
*L. laccata*	DAVFP:26723	Canada	JF899561	_	_	_	Direct Submission
*L. laccata*	Sporome	Mexico	KY969630	_	_	_	Direct Submission
*L. laccata* var. *pallidifolia*	HMJAU26932	China	KM246792	_	_	_	[20]
*L. laccata* var. *pallidifolia*	GMM7605	France	JX504146	KU685901	KU686154	KU686048	[21]
*L. laccata* var. *pallidifolia*	lac 1370	USA	DQ149849	_	_	_	[20]
*L. lateritia*	GMM7531	Australia	KU685709	KU685850	KU686118	KU685992	[1]
*L. lateritia*	GMM7221	Australia	KU685663	KU685806	_	KU685949	[1]
*L. lateritia*	GMM7250	Australia	KU685668	KU685811	_	KU685954	[1]
*L.* *longipes*	MQ 18R253-QFB30769	Canada	MN992191	_	_	_	Direct Submission
*L.* *longipes*	HMJAU26933	China	KM246793	_	_	_	[31]
*L.* *macrobasidia*	HBAU15557	China	MW871602	_	_	_	Direct Submission
*L.* *macrocystidia*	GMM7616	France	KM067850	KU685863		KU686004	[48]
*L.* *macrocystidia*	GMM7612	France	KM067841	KU685861	KU686122	KU686002	[48]
*L.* *macrocystidia*	GMM7626	France	KM067856	KU685865	KU686125	KU686006	[48]
*L.* *masoniae*	GMM7443	Australia	JX504139	JX504215	_	_	[21]
*L.* *masoniae*	GMM7240	Australia	KU685667	KU685810	_	KU685953	[21]
*L.* *masoniae*	GMM7200	Australia	KU685656	KU685799	KU686084	KU685941	[21]
***L.* ** ***miniata* ** **(T)**	**GDGM76043**	**Guangdong, China**	**OR689440**	**OR785476**	**_**	**_**	**This study**
*L.* *montana*	C5442	Switzerland	OR419936	_	_	_	Direct Submission
*L.* *montana*	M5464	Switzerland	OR419935	_	_	_	Direct Submission
*L.* *montana*	TWO 319	USA	DQ149862	_	_	_	[32]
*L. moshuijun* (T)	HKAS93732	Yunnan, China	KU962989	_	_	_	[32]
*L.* *moshuijun*	MB-001113	China	KU962985	_	_	_	[32]
*L. moshuijun*	HMAS 131870	China	ON877154	_	_	_	[32]
*L.* *murina*	Nara_LaM90	Japan	AB211271	_	_	_	[55]
*L.* *murina*	FBSR-535	Pakistan	OQ881021	_	_	_	[57]
*L.* *murina*	ASIS24249	Korea	MG519552	MG519592	MG551658	MG551625	[55]
***L.* ** ** *nanlingensis* **	**GDGM84949**	**Guangdong, China**	**OR689441**	**OR785477**	**OR826274**	**OR835198**	**This study**
***L.* ** ***nanlingensis* ** **(T)**	**GDGM84954**	**Guangdong, China**	**OR689442**	**OR785478**	**OR826273**	**OR835199**	**This study**
*L. negrimarginata* (T)	BAP360	Tibet, China	JX504120				[21])
*L. negrimarginata*	GMM7631	Tibet, China	JX504153	JX504227	KU686130	KU686011	[21]
*L. negrimarginata*	HMAS272239	Tibet, China	ON877189	_	_	_	[30]
***L.* ** ***neovinaceoavellanea* ** **(T)**	**GDGM52852**	**Jiangxi, China**	**OR689447**	**OR785479**	**_**	**_**	**This study**
** *L. neovinaceoavellanea* **	**GDGM53063**	**Jiangxi, China**	**OR689448**	**OR785480**	**_**	**_**	**This study**
** *L. neovinaceoavellanea* **	**GDGM89621**	**Yunnan, China**	**OR689449**	**OR785481**	**_**	**_**	**This study**
*L.* *nitrophila*	Corrales 467	Panama	MT279233	_	_	_	[51]
*L.* *nitrophila*	Corrales 595	Panama	MT279236	MT279211	MT436074	MT431186	[51]
*L.* *nitrophila*	Corrales 423	Panama	MT279235	_	_	_	Direct Submission
*L.* *nobilis*	nob1469	USA	DQ149854	_	_	_	[20]
*L.* *nobilis*	nob42527	USA	DQ149861	_	_	_	[20]
*L.* *nobilis*	F1091206	USA	KU685636	KU685779	_	_	[1]
*L.* *ohiensis*	GMM7564	New Zealand	KU685715	KU685856	_	KU685997	[1,21]
*L.* *ohiensis*	GMM7593NZ	New Zealand	KU685718	KU685860	KU686121	KU685994	[1]
*L.* *oblongospora*	OblFr	France	GQ406466	_	_	_	[49]
*L.* *ochropurpurea*	JMP0038	USA	EU819479	_	_	_	[58]
*L. ochropurpurea*	PRL3777	USA	JX504169	JX504246	_	KU686024	[21]
*L. ochropurpurea*	PRL4777	USA	KU685721	KU685883	_	KU686025	[21]
*L. pallidorosea* (T*)*	HKAS107730	Yunnan, China	MW540808	_	_	_	[2]
*L. pallidorosea*	HKAS53170	Yunnan, China	MW540809	_	_	_	[2]
*L*. *paraphysata*	PDD:80007	New Zealand	KM975424	_	_	_	Direct Submission
*L*. *paraphysata*	PDD:95230	New Zealand	KM975427	_	_	_	Direct Submission
*L. parva* (T)	SFC20120919-05	Korea	MG519529	MG519573	MG551640	MG551604	[59]
*L. parva*	SFC20121001-08	Korea	MG519530	MG519574	_	_	[59]
*L. parva*	SFC20120906-01	Korea	MG519527	MG519572	MG551639	MG551602	[59]
*L. populina* (T)	GDOR411	Italy	Mn871894	_	_	_	Direct Submission
*L. prava* (T)	HKAS106742	Guangdong, China	MN585660	_	_	_	[33]
*L. prava*	HKAS106745	Guangdong, China	MN585661	_	_	_	[33]
*L. proxima*	GMM7584	Russia	KU685717	KU685858	KU686120	KU685999	[1]
*L. proxima*	GMM7596	France	JX504142	JX504217	KU686151	KU686045	[1]
*L. proxima*	GMM7631	France	JX504152	JX504226	_	_	[1]
*L. pseudomontana*	Cripps 1771	USA	DQ149870	_	_	_	[20]
*L. pseudomontana* (T)	Cripps 1625	USA	DQ149871	_	_	_	[20]
*L. pumila*	GMM7637	France	JX504156	JX504229	KU686158	_	[21]
*L. pumila*	GG125_86	Netherlands	GU234161	_	_	_	[60]
*L. roseoalbescens*	LM5042	Mexico	KJ874327	KJ874330	_	_	[18]
*L. roseoalbescens*	LM5099	Mexico	KJ874328	KJ874331	_	_	[18]
*L. roseoalbescens*	VB4678	Mexico	KJ590509	KJ590510	_	_	[18]
*L. roseoalbescens*	VB4677	Mexico	KJ590508	KJ590511	_	_	[18]
*L. rubroalba*	HKAS90766	Yunnan, China	KX449359	_	_	_	[17]
*L. rubroalba*	HKAS90751	Yunnan, China	KX449360	_	_	_	[17]
*L. rubroalba*	HMAS131833	China	ON877153	_	_	_	[17]
***L. rufobrunnea* ** **(T)**	**GDGM82878**	**Yunnan, China**	**OR689443**	**OR785482**	**OR826272**	**OR835197**	**This study**
** *L. rufobrunnea* **	**GDGM89627**	**Yunnan, China**	**OR689444**	**OR785483**			**This study**
*L. salmonicolor* (T)	GMM7596tibet	Tibet, China	JX504143	JX504218	KU686151	KU686045	[21]
*L. salmonicolor*	GMM7602	Tibet, China	JX504145	JX504220	_	_	[21]
*L. squarrosa*	DM63	Mexico	MF669958	MF669965	_	_	[61]
*L. squarrosa*	DM121	Mexico	MF669960	MF669967	_	_	[61]
*L. squarrosa*	DM93	Mexico	MF669959	MF669966	_	_	[61]
*L. stellata*	SYC 207	Panama	KP877339	_	_	_	[62]
*L. stellata*	SYC 109	Panama	KP877340	_	_	_	[62]
*L. stellata*	Corrales 27	Panama	MT279231	MT279210	_	MT431185	[51]
*L. striatula*	HMJAU59796	China	OR468697	_	_	_	Direct Submission
*L. striatula*	1475 PREMIX	USA	OQ612526	_	_	_	Direct Submission
*L. striatula*	CNV105	USA	MT345281	_	_	_	Direct Submission
*L. tetraspora*	F1080957	Argentina	KU685631	KU685775	_	_	[1]
*L. tetraspora*	CT-4259	Argentina	MH930294	_	_	_	Direct Submission
*L. torosa*	SFC20150902-17	France	MG519561	_	_	_	[50]
*L. torosa*	KA12-1306	France	MG519562	_	_	_	[50]
*L. tortilis* (T)	ASIS22273	Korea	MG519533	_	_	_	[50]
*L. tortilis*	GMM7635	France	JX504155	KU685906	KU686156	KU686053	[21]
*‘L. tortilis’*	AWW545	USA	JX504106	JX504190	_	KU685917	
*‘L. tortilis’*	F1116205	USA	KU685641	KU685785	_	_	[1]
*L.* *trichodermophora*	GO-2010-082	Mexico	KC152152	_	_	_	Direct Submission
*L.* *trichodermophora*	HC-PNNT-099	Mexico	KT875034	_	_	_	Direct Submission
*L.* *trullisata*	PRL7587	Tibet, China	JX504170	JX504247	KU686153	KU686047	[21]
*L.* *trullisata*	WCG2075	–	KM067894	_	_	_	[48]
***L.* ** ** *umbilicata* **	**GDGM82883**	**Yunnan,China**	**OR689445**	**OR785485**	**OR826270**	**OR835194**	**This study**
***L.* ** ***umbilicata* ** **(T)**	**GDGM82911**	**Yunnan,China**	**OR689446**	**OR785486**	**OR826268**	**OR835192**	**This study**
*L. versiforma* (T)	SFC20120926-01	Korea	MG519556	MG519594	MG551660	MG551627	[50]
*L.* *versiforma*	ASIA20939	Korea	MG519557	MG519595	MG551661	MG551628	[50]
*L.* *versiforma*	KUN20120924-82	China	MG519559	MG519596	MG551662	MG551629	[50]
*L.* *violaceonigra*	GMM7520	New Zealand	KU685707	KU685848	_	KU685990	[1]
*L.* *violaceonigra*	GMM7580	New Zealand	KU685716	KU685857	_	KU685998	[1]
*L.* *violaceonigra*	GMM7533	New Zealand	KU685710	KU68585	_	KU685993	[1]
*L.* *vinaceoavellanea*	SFC201200907-18	Korea	MG519536	MG519578	_	MG551611	[50]
*L.* *vinaceoavellanea*	ASIS23860	Korea	MG519538	_	MG551645	MG551613	[50]
*L.* *vinaceoavellanea*	SFC20120907-10	Korea	MG519539	MG519580	MG551646	MG551614	[50]
*L.* *vinaceoavellanea*	TNS A2986	Korea	JN942810	_	_	JN993520	Direct Submission
*L.* *vinaceobrunnea*	GO-2009-360	Mexico	KC152154	_	_	_	Direct Submission
*L.* *vinaceobrunnea*	GO-2009-316	Mexico	KC152155	_	_	_	Direct Submission
*L.* *vinaceobrunnea*	GO-2009-308	Mexico	KC152153	_	_	_	Direct Submission
*L. yunnanensis* (T)	KUN-F78558	Yunnan, China	JQ670897	_	_	_	[22]
*L. yunnanensis*	HMAS271371	Yunnan, China	ON877187	_	_	_	[22]
*L. yunnanensis*	HMAS264310	China	KX496978	_	_	_	Direct Submission
*Mythicomyces corneipes*	AFTOL972	Germany	DQ404393	AY745707	DQ029197	DQ447929	Direct Submission
*M.* *corneipes*	DAOM178138	Germany	_	_	_	_	Direct Submission

**Table 2 jof-09-01179-t002:** A morphological comparison of species in *Laccaria* described from China.

Species	Habit	Pileus Diameter	Pileus Color	Lamellae Color	Spores Size	References
*L. acanthospora*	On sandy banks in mixed temperate alpine forest. Distributed in Tibet.	4–15 mm	Orange	Light orange	7–10 × 7–10 µm (av. 8.3–8.4 × 8.9–9.2 µm); spines 2–6 µm long	[21]
*L. alba*	In mixed forests with *Abies, Betula, Fraxinus, Picea, Pinus, Quercus, Tilia* and *Ulmus*. Distributed in Yunnan.	7–32 mm	Reddish brown to orange, fading to buff	Pink-salmon	7–10 × 7–11 µm (av. 8.5 × 8.6 µm); spines 1–5 (–6) µm	[21]
*L.* *angustilamella*	In forests dominated by *Quercus* and *Lithocarpus*. Distributed in Yunnan.	20–30 mm	Pinkish flesh-colored, slightly darker on radial striations, fading to buff with age or on drying	Pinkish	(8)8.5–11.5(12) × (7.5)8–11 µm (av. 9.2 ± 0.8 × 8.6 ± 0.8 µm); spines (2.0) 2.5–5 µm long	[30]
*L.* *anthracina*	In mixed forest, near Pinus wallichiana. Distributed in Tibet.	25–65 mm	Dark grayish brown, dark brownish gray to brownish black, fading to grayish brow after drying	Pinkish brown	(6.5)7–9.5(11) × (6.5)7–9(10.5) μm; spines 1.5–2.5 μm long	[34]
*L.* *aurantia*	In mixed broad-leaved forests dominated by *Quercus* and *Lithocarpus*. Distributed in Yunnan.	35–40 mm	Vividly orange	Orange	(8)9–10 (11) × (7) 8–10 μm; spines (0.75) 1–1.5 (1.75) μm	[22]
*L.* *bullipellis*	In mixed temperate alpine conifer forest with *Abies, Picea, Pinus, Rhododendron, Sorbus* and *Gamblea*. Distributed in Tibet.	22 mm	Brown to orange-brown	Brown to orange-brown	6–9 × 6–10 µm (av. 8.3 µm); spines 1–2 µm long	[21]
*L.* *fagacicola*	In subtropical broad-leaved forests with trees of Fagaceae. Distributed in Yunnan.	20–45 mm	Brownish orange to brownish	Brownish orange	(6.5)7–9(10) × 6.5–8(9) μm, (av. 7.7 ± 0.6 × 7.4 ± 0.5); spines 1–2(2.5) μm long	[2]
*L. fengkaiensis*	In broad-leaved forests dominated by Fagaceae, such as *Lithocarpus haipinii, L. litseifolius, L. glaucus, L. uvariifolius, Castanopsis fissa*, and *C. faberi*. Distributed in Guangdong.	50–90 mm	Orange-white, pale orange when young, light orange to pale red, pastel red with age	Pastel red to grayish red	5.2–6.3 × 5.1–6.3 μm; spines 0.5–1.2 μm long	[33]
*L.* *fulvogrisea*	In mixed broad-leaved forests with *Quercus, Eucalyptus, Lithocarpus* and *Ficus*. Distributed in Yunnan.	<30 mm	Gray with a violet to reddish brown tinge	Initially whitish, grayish to brownish with age	(7.5) 8–10 × 8–11 (7.5) μm; spines (1.5) 1.7–2.5 μm long	[22]
*L. himalayensis*	In mixed temperate alpine conifer forest with *Abies, Acer, Larix, Pinus* and *Salix*. Distributed in Tibet.	6–34 mm	Brown at disk, orange-pink toward the margin	Orange-pink	6.5–10 µm (av. 8.1–9 µm); spines (0.5–) 1–3 µm	[21]
*L.* *miniata*	In subtropical mixed forests with trees of Fagaceae (*Castanopsis fissa, C.* spp.) and Pinaceae (*Pinus massoniana*). Distributed in Guangdong.	10–15 mm	Red at mass, with a deep red center, fading to reddish orange to yellowish red toward the margin	Pastel red, pastel pink to red	8–10.5(11) × 8–10 μm, (av. 9.37 ± 0.83 × 8.8 ± 0.6); spines 0.5–1 μm long	Present study
*L.* *moshuijun*	In subtropical mixed forests with *Piuns yunnanensis, Cunninghamia lanceolata* and other broad-leaved trees. Distributed in Guizhou.	Up to 30 mm	Violet to bluish with a brown tinge at the center	Deep violet	(7.5)8–9 × (8)9–10 μm; spines 1-1.5 μm	[32]
*L.* *nanlingensis*	In subtropical, broad-leaved forests mainly dominated by Fagaceae trees. Distributed in Guangdong.	30–55 mm	Orange, reddish orange, orange-red, brownish orange to brownish red, fading to light orange to pale orange when dry	Concolorous with pileus or darker to pale red to grayish red	6.5–7.5 × 6–7 μm, av. 6.95 ± 0.15 × 6.37 ± 0.45; spines 0.5–1 μm long.	Present study
*L.* *negrimarginata*	In mixed temperate alpine conifer forest with *Abies, Acer, Larix, Pinus, Quercus* and *Salix*. Distributed in Tibet.	5–15 mm	Orange-brown fading to buff; squamules dark blackish brown to dark brown.	Pale pink to pinkish Gray	7–10 × 6–10 µm (av. 7.8–9 × 7.4–8.9 µm); spines up to 2 µm	[21]
*L.* *pallidorosea*	In subtropical broad-leaved forests with trees of Fagaceae. Distributed in Yunnan.	10–25 mm	Brownish to pinkish at center, becoming cream to white towards margin	White to pinkish	(6)7–9(10) × (6)6.5–8.5(9) μm, (av. 8 ± 0.7 × 7.75 ± 0.6); spines 1.5–2 (3) μm long	[2]
*L.* *prava*	In broad-leaved forests dominated by Fagaceae, such as *Lithocarpus glaber, L. corneus, Cyclobalanopsis pachyloma, Castanopsis faberi* and *C. fissa*. Distributed in Guangdong.	30–75 mm	Pastel red, pale red to reddish white, even to white with age or on drying.	Reddish white to grayish red	6.5–7.5 × 7–8 μm; spines 0.5–1 μm long	[33]
*L.* *rubroalba*	In a tropical forest dominated by Fagaceae and other broad-leaf trees. Distributed in Yunnan.	22–40 mm	Reddish white when moist or young, becoming white to paler when dry	Flesh-colored	(5) 6–9 (10) × 5–7 (8) μm; spines 1.2–2.7 μm long	[17]
*L.* *rufobrunnea*	In subtropical mixed forests mainly dominated by Fagaceae trees and pine trees. Distributed in Yunnan.	12–35 mm	Brownish orange to brownish red, usually fading to brownish yellow in dry condition	Pastel red, pink to purplish pink, usually changing pinkish white to purplish white in dry condition	8–9 × 7–8 μm, av. 8.15 ± 0.23 × 7.57 ±0.5; spines 1–2 μm long	Present study
*L.* *salmonicolor*	In mixed temperate alpine forests with *Betula, Larix* and *Picea*. Distributed in Tibet.	10–35 mm	Reddish brown, to pale brown-buff	Pinkish salmon	7.5–10 µm (av. 8.5 µm); spines 1–3 µm long	[21]
*L.* *umbilicata*	In subtropical mixed forests mainly dominated by *Pinus yunnanensis*, mixed with a small number of Fagaceae trees. Distributed in Yunnan.	10–28 mm	Pale yellow, pale orange to light orange, changing light brown to brown when in dry condition	Orange-white, to pinkish white, sometimes pale orange to pastel red in dry conditions	(7)8–10 × (7)8–10 μm, av. 9.03 ± 0.75 × 8.67 ± 0.82; spines 1.5–2 (2.5) μm long	Present study
*L. neovinaceoavellanea*	In subtropical broad-leaved forests dominated by Fagaceae. Distributed in Jiangxi.	15–40 mm	Pastel pink, rose, purplish pink to pale violet at mass, grayish magenta to dull violet at center	Concolorous with pileus or paler	7–8 × 7–8 μm, av. 7.6 ± 0.4 × 7.35 ± 0.42; spines 1–2 μm long	Present study
*L.* *yunnanensis*	In subtropical forests. Distributed in Yunnan.	60–100 mm	Brownish to flesh-colored	Flesh-colored	(7.5)8–9 × (7.5)8–10 μm; spines 1–1.5 μm long	[22]

## Data Availability

The datasets presented in this study can be found in online repositories. The names of the repository/repositories and accession number(s) can be found below https://www.ncbi.nlm.nih.gov/genbank/, and https://nmdc.cn/fungalnames/. (accessed on 1 November 2023).

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
