# Peer review of "Morphology and Molecular Phylogeny Reveal Five New Species of Laccaria (Hydnangiaceae, Agaricales) from Southern China"

_jof, 2023, doi:10.3390/jof9121179_

Round 1

Reviewer 1 Report

Comments and Suggestions for Authors

Five species of Laccaria from southern China, new to science, were proposed based on morphological and phylogenetic studies. Morphological studies were performed using classical methods. Their results provide a good idea of the features of the fruiting bodies. The phylogeny is based on a multilocus approach. The sequences of the new species are grouped into well-supported clades; their genetic isolation is beyond doubt. Five species new to science represent a significant contribution to the knowledge of the mycobiota of Southeast Asia and significantly complement the understanding of the genus Laccaria as a whole. The work is well structured, based on significant primary material and provided with comprehensive illustrations. Small corrections of the text are needed. See the attached file.

Author Response

Dear reviewer,
Thank you very much for taking the time to review our manuscript. According to your suggestions, we have made some improvements, please find the corresponding revisions in track changes in the re-submitted files.

Thank you again.

Best wishes,

Ming ZHANG

Reviewer 2 Report

Comments and Suggestions for Authors

Grammatical corrections:

Line 47.  do you mean 15 species originally described from China (not just reported)
50. five species are confirmed (present tense)
51. are formally . . . 
59. codes follow Kornerup . . .
118. ie not needed: new species, L. neovin . . . . .
141. replace  differs with is distinguished 
152. subtomentose 
         what does at mass mean here (and elsewhere)? 
185. replace showed with shows; miniata is close to . .  .
191, 193. correct spelling of brown
196. delete which
202. replace larger with large
214. delete with
222. provide the spine length
244. larger basidiocarps 
246. replace besides with In addition
247. delete to. 
        replace comman with semicolon: nanlingensis; however . . . 
248. basidiocarps
254. spelling of lamellae
321. italicize Laccaria
330. delete with
365. spelling of lamellae 
385. delete with
431. reword: Previous studies based on morphological . .  . 
447. delete of and with 
454. reword: In addition, basidia size and whether they are 2 or 4-spored can be useful distinguishing features. Most species of  . . .  (new sentence)
459-465. Sentence beginning with Laccaria miniata and  . . .  and ending with  . . . Fagaceae trees is too long and should be divided into several sentences or clauses (semicolons)
466. delete which; change were to are 

Other comments:

Several references were spot-checked.  They are cited in the references but could not be found in the text: 67, 77, 72.  

Two species are in bold in the table - nanlingensis and rufobrunnea.  are the other newly described species supposed to be in bold also? 

The taxa in the table are in alphabetical order and reading it is cumbersome.  Would it be possible to group related taxa or morphologically similar taxa together, or by habitat or geographic area, whichever works best?  This might convey comparison information better to the reader.  A dichotomous key might be recommended but may be difficult because some of the morphological features are so similar.  

This reviewer is not familiar with the climate/ecology of China.  It is recommended that in the Methods and Materials, a little more information be provided about the "subtropical" area where the specimens were collected.  Can lat/long coordinates be provided?

Laccaria is a diverse complex genus.  The authors have done an excellent job of presenting their research.  

Comments on the Quality of English Language

Grammar/spelling corrections are indicated in the above comments.  The English is generally excellent.  Most corrections are minor. 

Author Response

Dear reviewer,
Thank you very much for taking the time to review our manuscript. According to your suggestions, some improvements have been made in the text, please find the corresponding revisions in track changes in the re-submitted files.
Besides, several references were not found in the text, because they were cited in the supplymentary file.

Thank you again,

Best wishes

Ming ZHANG
